# Biocorrosion of Carbon Steel under Controlled Laboratory Conditions

**Francisco Córdoba** [1,*] and **Aguasanta M. Sarmiento** [2]

1 Department of Integrated Sciences, University of Huelva, 21007 Huelva, Spain
2 Department of Mining, Mechanical, Energy, and Construction Engineering, University of Huelva, 21007 Huelva, Spain; amsarmiento@uhu.es
* Correspondence: fcordoba@uhu.es

**Abstract:** In the Iberian Pyritic Belt (SW Europe), Acid Mine Drainage (AMD) is the consequence of the interaction of physical-chemical and biological factors, where aerobic Fe and/or S oxidizing chemolithotrophic and anaerobic sulfate reducing bacteria play an essential role. As a result, the polluted waters are highly acidic (pH 2–3) and contain numerous dissolved or suspended metals, which gives them a powerful corrosive action on constructions related to mining activities with high economic losses. To verify the role of bacteria in the corrosion of carbon steel, a common material in buildings exposed to corrosion in acidic waters, several experiments have been carried out under controlled conditions using carbon steel bars and acidic water containing bacteria consortia from an AMD river of the Iberian Pyritic Belt. In all the experiments carried out, a remarkable oxidation of supplemented iron was observed in the presence of bacteria. Using carbon steel as the sole iron source, we observed a slight corrosion of the bars, but when culture media was supplemented with elemental sulfur, steel bars was severely damaged. Since the bacteria inoculum come from the surface water, well oxygenated, nutrient-poor river, the obtained results are discussed based on facultative metabolism of acidophilic chemolithotrophic bacteria.

**Keywords:** AMD; biocorrosion; carbon steel; acidophilic bacteria; Iberian Pyritic Belt

## 1. Introduction

The Iberian Pyrite Belt (IPB) probably represents the largest accumulation of poly-metallic massive sulfides on Earth's crust. It is approximately 350 million years old. It extends for about 250 km and the average width is about 30 km. The history of its exploitation dates back over 4000 years in which copper, silver, gold, and more recently sulfur (of interest for manufacturing sulfuric acid) have been the most demanded metals [1,2]. Therefore, it is one of the oldest mining districts in the world with continuous activity, mainly since the mid-19th century. However, the ancient exploitation procedures have generated an environmental liability of 90 mines, 700 hectares of open pits, 300 hectares of mine waste dumps, 860 hectares of tailings ponds, and 280 hectares of other facilities [3].

The exposure of pyrites to atmospheric oxygen and water causes their oxidation, generating acidity, sulfates, and various oxidized forms of iron. The runoff waters flowing and passing through the materials of the Pyrite Belt acquire a characteristic reddish color and acidity, while carrying with them a huge amount of leached metals through the river basins of the Tinto and Odiel rivers to the Atlantic Ocean. This naturally occurring phenomenon is called Acid Rock Drainage (ARD). When there is an anthropogenic factor derived from the mining of massive sulfides, the term Acid Mine Drainage (AMD) is used.

Acid Mine Drainage (AMD) is a set of natural processes of mineral oxidation and leaching of massive sulfides, derived from mining and industrial activities. All of this explains the extreme acidity of the rivers and the suspended transport of metals derived from pyrite, chalcopyrite, sphalerite, among others [4]. The AMD generated in the IPB represents one of the largest environmental impacts and degradation in Europe [4–6].

Given the magnitude of the environmental problem posed by AMD, due to the level of acidity and high concentration of heavy metals transported by the affected rivers, it is considered necessary to restore the affected soils and waters using essentially passive treatments, due to their ease of implementation, relatively low cost, and low maintenance [7].

The waters of the Tinto River contain a rich community of bacterial -prokaryotic- and eukaryotic species, including more than 800 species of algae, fungi, and protozoa [8,9]. These studies have been confirmed and expanded for similar environments by other authors [10,11]. In a river affected by AMD, the diversity and relationships between different groups of bacteria are very complex. The oxygen gradient conditions the bacterial groups that occupy this ecological niche. The most abundant bacteria found in the most oxygenated zones are classified, from a metabolic point of view, as aerobic chemoautotrophic. This terminology indicates that their source of energy and reducing power comes from reduced inorganic matter, derived from massive sulfides. The optimal pH for the growth and functioning of these bacteria is around 2, so they are called extreme hyperacidophilic bacteria. This group includes iron and/or sulfur-oxidizing bacteria, as *Acidithiobacillus ferrooxidans*, *Leptospirillum ferrooxidans* as well as heterotrophic bacteria as *Acidiphilium* sp, which explain the high content of ferric iron in the water ("red rivers") and the high concentration of sulfates.

In the less oxygenated zones, such as the bottom and sediments of the rivers, anaerobic bacteria develop that reduce sulfates using low-molecular-weight organic compounds as reducing agents. Their activity generates sulfides ($H_2S$ and metallic sulfides). The most important group of these microorganisms are called sulfate-reducing bacteria (SRB) [12,13].

*A. ferrooxidans* is considered the main responsible for AMD due to its capacity to oxidize iron and sulfur (*L. ferrooxidans* cannot oxidize sulfur) and acidify the waters. In addition to fixing $CO_2$, it can reduce and fix $N_2$. Its optimal pH for growth is 2.0 [14,15]. Moreover, its metabolic versatility is extraordinary as it has a facultative metabolism: under aerobic conditions, its metabolism is oxidative, however, under anaerobic conditions, it reduces sulfates generating sulfides, it can use $H_2$ as an electron donor, and even reduce $Fe^{3+}$ [16].

In AMD environments, bacteria involved in the sulfur cycle are implicated in biocorrosion processes of construction materials. These bacteria include both sulfur-oxidizing and sulfur-reducing bacteria. They form biofilms on the mineral's metallic surface, making their removal difficult. This creates highly aggressive and dangerous conditions for mining industries or infrastructure as they irreversibly destroy the mineral [17–19].

For this reason, the consequences of biocorrosion in these environments are an issue of industrial and economic interest, whose mechanisms should be identified in order to propose protection and repair systems, if necessary.

The aim of this work has been to verify and measure the biocorrosive activity of bacterial consortia collected from waters affected by AMD on carbon steel test tubes, using steel as the sole source of iron. The effects of the addition of elemental sulfur to the culture media have also been evaluated. Interestingly, the addition of sulfur to the culture media allowed us to observe a significant change in the dynamics observed in its absence, as the initial oxidative metabolism combined with a reducing metabolism, which explains the observed oscillations in pH, conductivity, and redox potential, as well as the changes in the appearance of precipitates.

## 2. Materials and Methods

In all experiments, the bacteria used were sourced from well-oxygenated surface waters of the Agustin River (37°32′48.2″ N, 7°04′56.0″ W), one of the rivers that make up the Odiel River basin. The Agustin River originates south of the Tharsis mines and is heavily affected by AMD, with a pH of 1.8–2.2, a conductivity greater than 4 mS/cm, and a highly oxidizing redox potential (Eh ≈ 900 mV) and no detectable organic material.

In the laboratory, the collected bacteria were cultured in 9K medium using ferrous sulfate (44.4 g/L) as the source of iron [19]. Cultures were carried out in in 100 mL Erlenmeyer flasks using an orbital shaker (100 rpm, 30 °C). The culture medium was renewed

weekly to allow bacterial cultures to adapt to laboratory conditions. The proportion of oxidized and reduced iron was measured in the media containing iron salts using the 1,10-phenanthroline method [20]. After 4 weeks of culture, inocula of the adapted culture were added to 9K medium (10% *v/v*) in the absence of iron salts, which were replaced by carbon steel test tubes ($60 \times 10 \times 0.8$ mm$^3$; $10 \pm 1$ g). The chemical element composition (wt%) of S-235JR carbon steel tubes was: 0.067 C, 0.53 Mn, 0.008 Si, 0.016 P, 0.013 S, 0.031 Al and Fe balance (S/EN 10204) (Data provided by the supplier).

The bacteria were incubated using steel as the sole source of iron for 30–45 days. Elemental sulfur was added to the culture medium (2% *w/v*) when indicated. The weight variation of the carbon steel test tubes was periodically recorded. Changes in pH, conductivity (EC), and redox potential (ORP) were measured with a multiparameter analyzer (Hanna HI 9828, HANNA, Woonsocket, RI, USA). Color changes were measured at a wavelength of 510 nm using a spectrophotometer. The presence of bacteria in the culture medium was verified by fresh observation under a Nikon Eclipse E400 contrast phase microscope (Nikon, Tokyo, Japan). In all experiments, and in order to analyze whether the observed effects were due to the presence of bacteria or to the chemical composition of the culture medium, the bacterial inoculum was filtered using sterile 0.20 μm filters to ensure the absence of bacteria.

## 3. Results

### 3.1. Control Experiments: Oxidation of Ferrous Salts

First, the oxidizing capacity of the bacteria collected from water contaminated by AMD was tested by culturing them in 9K medium + FeSO$_4$ as a source of iron. The results indicated that after only 3 days of incubation, the medium containing bacteria acquired a conspicuous reddish color, as around 80% of the added iron was oxidized. In the absence of bacteria, the percentage of oxidized iron was less than 10% (Figure 1).

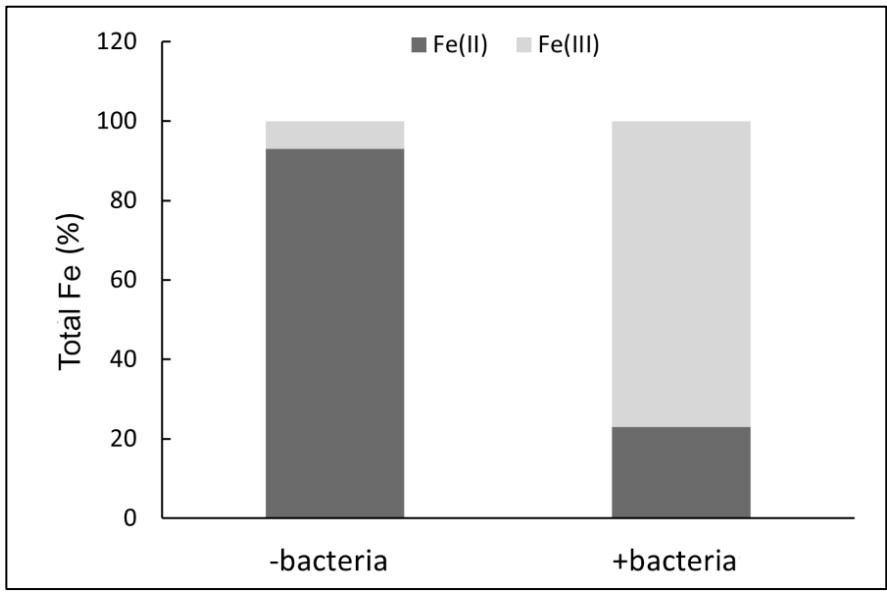

**Figure 1.** Oxidation of ferrous salts by bacteria from an AMD−polluted river. Ferrous and ferric ions were measured by the phenantroline method in 9K + FeSO$_4$ media cultures of 3 days incubation in the presence of a 10% bacterial inoculum (+bacteria) or using a 0.20 μm filtered inoculum (−bacteria).

### 3.2. Using Carbon Steel as the Sole Source of Iron

In a new experiment, a series of carbon steel test tubes were used as a nutritional source of iron for aerobic bacteria obtained from the Agustín River. As in previous cases, a control was performed with sterile filtered inocula. 6 flasks were used: 3 without added bacteria

(controls) and 3 with steel as a nutritional source. The experiment lasted for 32 days. Before starting the experiment, the steel bars were sterilized in an autoclave.

The results obtained show that, in all cases, numerous precipitates were produced at the bottom of the flasks, probably iron oxyhydroxides, which were more abundant in the flasks with bacteria. The steel test tubes also had attached precipitates that, due to their appearance and color, appear to be, as before, iron oxyhydroxides. Once cleaned of the precipitates, the steel test tubes did not appear to have any appreciable differences visually. The test tubes barely lost any weight from the beginning of the experiment in the absence of bacteria, since the weight decreased by 1 %. However, in the presence of bacteria the decrease in the weight of steel tubes was 9%.

The color variation of the medium was also measured during the course of the experiment, since the production of oxidized iron was expected to color the media. To do this, the media was filtered with filter paper (to remove the precipitates) and the absorption at 450 nm was measured on a spectrophotometer. The results show that, in the absence of bacteria, the media remained virtually colorless, while in the presence of bacteria, the media was tinted with yellowish hues, a clear indication of the oxidation of the iron contained in the test tubes and, if applicable, the iron sulfate added as a supplement.

During the incubation period, various changes in the physicochemical parameters of pH, conductivity, and redox potential occurred (Figure 2). Note that the pH and conductivity decreased in the media with the presence of bacteria. In their absence, the pH increased from 2.2 to 3.4, while there were no changes in conductivity. With respect to the redox potential, its continuous increase in the presence of bacteria was notable, while in the control media (without bacteria), the potential initially decreased, although it subsequently remained constant. In the same figure, a graph has been added that shows the inverse relationship between pH and redox potential in media with bacteria.

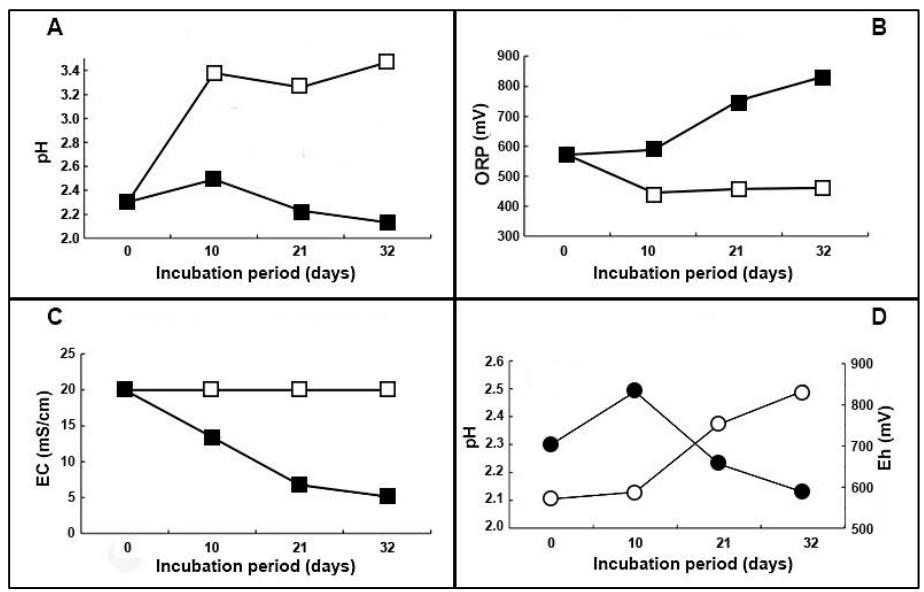

**Figure 2.** Changes in physico-chemical parameters during the incubation period of steel samples. (**A**) pH, (**B**) redox potential, (**C**) conductivity. White square: sterile inoculum (without bacteria); black square: unfiltered inoculum (with bacteria). In (**D**) the inverse relationship between pH and redox potential is shown). Black circles indicate pH and white circles the ORP.

### 3.3. Using Carbon Steel as the Sole Source of Iron and Elemental Sulfur

Since the steel test tubes barely contained sulfur (<0.03%), a new experiment was carried out, whose design was very similar to the one reported previously, but adding 1 g of elemental sulfur to the incubation media, given the ability of *Acidithiobacillus ferrooxidans* to oxidize sulfur to sulfate under aerobic conditions. As in the previous case, as the incubation time passed, the medium acquired color due to iron oxidation, especially in the presence of

bacteria. However, a new effect was palpable: the darkening of the medium and the strong smell of hydrogen sulfide in the final stages of incubation, which implies an unexpectedly prior reduction of sulfur to sulfide.

Numerous precipitates were also produced adhering to the test tubes. Although with very different qualities between the test tubes incubated in the absence of inoculated bacteria and those that remained in media containing bacteria from the Agustin River. In the former case, the precipitates were yellowish in color (probable iron oxihydroxides), while in the latter case the precipitates were of two types: yellowish and blackish (probable metallic sulfides).

Regarding the physicochemical parameters, multimodal profiles were observed in the media containing inoculated bacteria, with rises and falls in the observed values, and in any case with profiles very different from those observed in the previous experiment without sulfur. Again, it should be noted that the pH curve and the redox potential curve are approximately inverse. Regarding the control (in the absence of inoculated bacteria), the profiles show a continuous increase in pH, a decrease in redox potential, and no changes in conductivity (Figure 3). In other words, profiles very similar to those observed in the absence of sulfur (Figure 2). Note again the inverse relationship between the variation of pH and the redox potential.

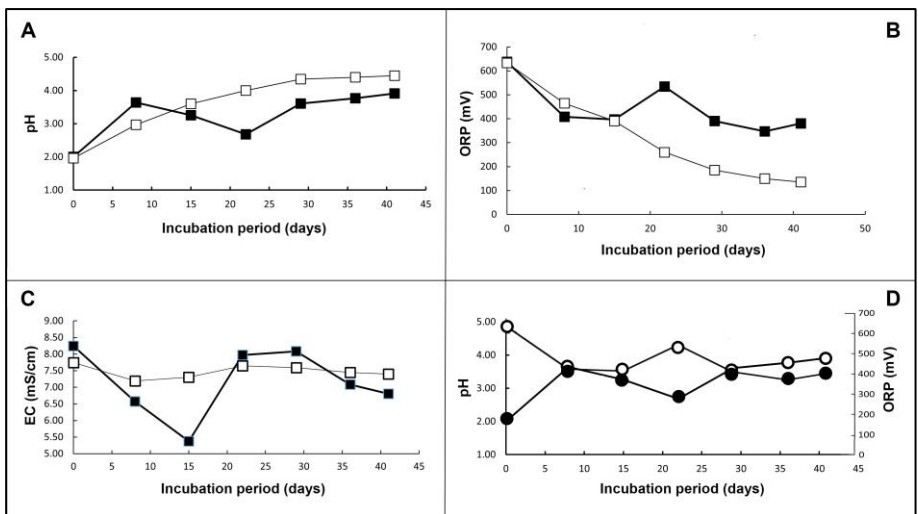

**Figure 3.** Changes in physico-chemical parameters during the incubation period of steel samples in sulfur-enriched cultures. (**A**) pH, (**B**) redox potential, (**C**) conductivity. White square: sterile inoculum (without bacteria); black square: unfiltered inoculum (with bacteria). In (**D**) the inverse relationship between pH and redox potential is shown). Black circles indicate pH and white circles the ORP.

In this case, in the steel samples incubated in the presence of the bacterial inoculum and sulfur, including numerous bacteria (Figure 4A), there was a notable weight decrease (Figure 4B) and, remarkably, conspicuous levels of corrosion in the samples exposed to the inoculated bacterial consortia (Figure 4C). Both steel tubes (incubated with and without bacteria) contains conspicuous but different colour precipitates: yellow in the absence of bacteria and mainly black with some yellow spots in the presence of bacteria. No bacteria were observed in the cultures using sterilized inoculum. The precipitates were carefully removed with pressurized water to observe the texture of the test tubes.

This corrosion was barely visible in the samples incubated in the absence of bacteria (sterile inoculants). The photos shown were taken after removing the precipitates adhered to the samples.

Through phase contrast optical microscopy, it was confirmed that only the media inoculated with unfiltered water from the Rivera de Agustin contained bacteria. In contrast, media inoculated with sterile (filtered) water did not contain bacteria.

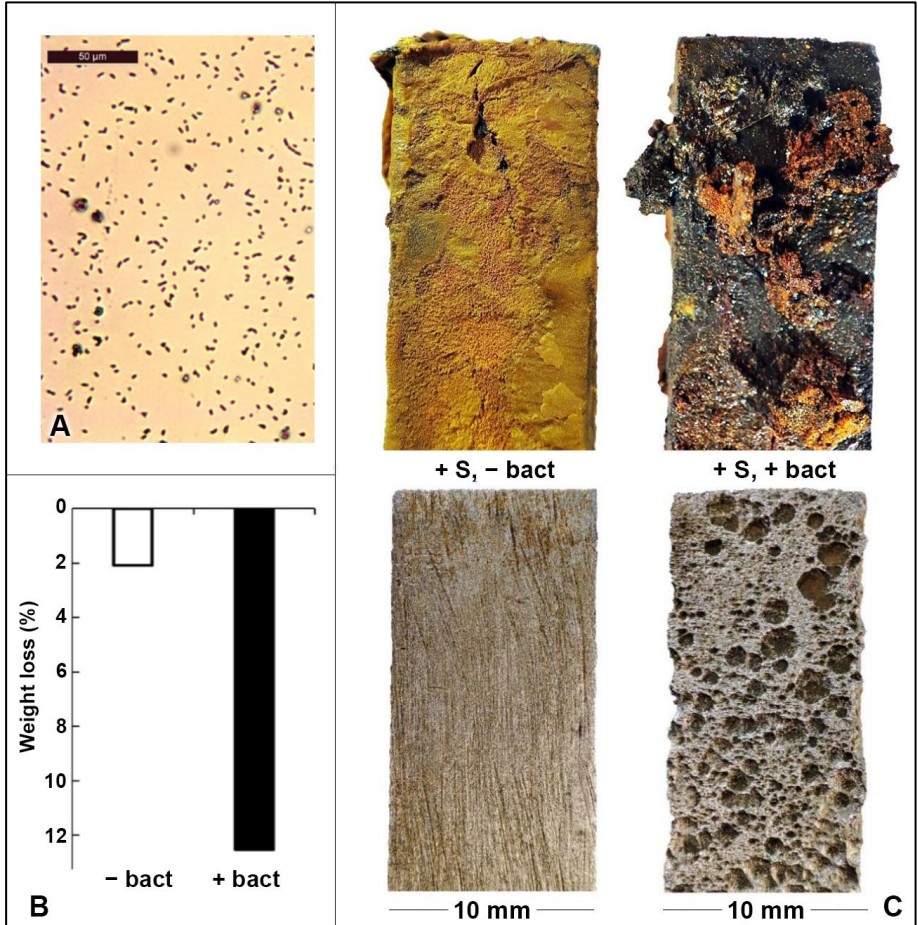

**Figure 4.** Changes in steel test tubes in the presence of bacteria and sulfur. (**A**) Bacteria as seen under a microscope (50 μm scale). (**B**) Variation in weight of steel samples in the absence of inoculated bacteria (inoculum sterilized by filtration) and in the presence of bacterial consortia. (**C**) Photographs of steel samples after incubation in media without bacteria (**left**) and in the presence of bacteria (**right**). Above: precipitates over steel samples. Below, steel test tubes after removal of precipitates.

Overall, the data from this last experiment clearly indicate that steel biocorrosion only occurred in media incubated in the presence of bacteria. However, and unlike the previous experiment, the presence of sulfur generated much more intense corrosion. In addition, the irregular profile of the evolution of physical-chemical parameters, as well as the simultaneous presence of yellowish precipitates (probably iron oxihydroxides) and blackish ones (probable metallic sulfides), in addition to the strong smell of hydrogen sulfide, allow us to suggest that both oxidation and reduction processes occur simultaneously.

As a control, an experiment was conducted under the same agitation conditions as the previous experiment, where bacterial inocula were cultured exclusively with sulfur without adding any source of iron. The results showed that sulfur alone was not able to support any type of bacterial growth In another control experiment, the same culture medium (9K + Fe + S + bacterial inoculum) was used, although in flasks strongly agitated by a magnetic bar at their base, placed on a thermostat-controlled magnetic stirrer. This system effectively mixes the volume of liquid from bottom to top, creating vortices that facilitate medium oxygenation. In this case, the medium acquired the characteristic color tones of iron oxidation, but did not produce any $H_2S$ odor or blackish precipitates (results not shown).

## 4. Discussion

AMD is a dramatic natural phenomenon of water pollution in mining environments where polymetallic sulfides accumulate, as is the case in the Iberian Pyritic Belt. Mining activities developed primarily from 1850 until the last third of the 20th century caused an accumulation of mining waste, where the mineral is highly fragmented and therefore offers a very high surface/volume ratio. These residues, exposed to the atmosphere, are slowly oxidized by oxygen while the acidity increases. When the pH is below 4, the oxidized iron produced acts as a catalyst, replacing the oxygen, while a variety of aerobic iron- and/or sulfur-oxidizing bacteria greatly increase the rate of oxidation of metallic sulfides and produce more protons [21,22]. The result is the release of a variety of bioleached metals into acidic waters (pH 2–3), which are transported dissolved or in suspension to the mouths of the affected rivers, Tinto and Odiel, in the Atlantic Ocean.

These AMD-polluted mining areas contain numerous installations associated with the mining industry, which under the influence of contaminated waters undergo processes of corrosion with significant economic repercussions but also affecting the safety of mining workers. In general, the most common materials used in these constructions are steel and concrete.

Although the microbiology of acidic waters in rivers contaminated by AMD is very complex, the mechanisms of biocorrosion are usually explained by the oxidation of iron and sulfur caused by aerobic chemolithotrophic bacteria, such as *Acidithiobacillus ferrooxidans* and *Leptospirillum ferrooxidans*, or by the production of $H_2S$ catalyzed by the action of anaerobic heterotrophic sulfate-reducing bacteria. However, these explanations may be an oversimplification of the processes that occur in waters contaminated by AMD, since there are numerous bacterial species that form metabolic consortia, and some of these species may be facultative, such as *Ac. ferrooxidans*.

In this paper, we have focused on the preliminary analysis of carbon steel biocorrosion under controlled laboratory conditions. The first step consisted of testing the iron oxidation capacity of bacterial consortia collected from a river affected by AMD. To do this, a volume of water from the contaminated river was added to 9K medium supplemented with $FeSO_4$ (44.4 g/L) and with an initial pH value of $2.0 \pm 0.2$.

The results clearly indicate that bacteria are involved in the oxidation of added ferrous iron since at pH < 3.5 pyrite oxidation by $Fe^{+3}$ proceeds at a much faster rate than oxidation by dissolved oxygen. It has been demonstrated that the abiotic rate of ferrous iron oxidation is the slowest reaction in the AMD iron system at pH < 4. This reaction is unable to produce $Fe^{+3}$ iron as fast as pyrite can consume it. In contrast, microbes dramatically increase the rate of this reaction by as much as six orders of magnitude, depending on pH [22]. The microbial oxidation of dissolved $Fe^{2+}$ to $Fe^{3+}$ is a key to understanding the oxidation of sulfide minerals because the redox potential of the ferrous/ferric iron couple is related to pH and is most positive in extremely acidic environments (Eh $\frac{1}{4}$ +770 mV at pH 2), implying that ferric iron is an attractive alternative electron acceptor to oxygen in low pH environment [23–25]. On the other hand, it cannot be ignored that the inoculum used contains oxidized iron as it comes from waters affected by AMD from the Agustin River. In this regard, it should be considered that the Agustin River contains an average concentration of 243 mg/L of iron [26], largely oxidized (hence the intense red color of this river). However, this average value is highly variable, depending mainly on the rainfall. Specifically, the iron concentration in the sample used in this study was as high as 1543 mg/L, with over 90% in oxidized form. The recorded in situ values of pH, EC, and Eh' were 2.3, 3.7 mS/cm, and 890 mV, respectively. Considering this value, the 5 mL added to the medium would represent 7.5 mg of iron (practically negligible compared to the 44.4 g of iron per liter of 9K medium added to the medium as a nutritional source used for the culture enrichment of *A. ferrooxidans* [19]).

This former experiment also represents a simulated AMD, where it is demonstrated that in the absence of bacteria, this dramatic form of contamination would hardly be

significant. In other words: AMD can only be explained if the presence of iron and sulfur oxidizing bacteria is considered.

This type of culture allowed for the growth of bacterial species that were used as inoculum to test their effect on carbon steel test tubes. In this new round of experiments, the culture medium lacked a specific reduced iron source as the ferrous sulfide salts, which were replaced by steel.

After four weeks of cultivation, conspicuous changes were observed in the liquid culture medium. The pH decreased in the presence of bacteria, due to the proton production associated with the so-called hydrolysis of ferric iron. The conductivity probably decreased due to the precipitation of iron oxyhydroxide salts (similar to schwertmannite), which could be observed with the naked eye, while the redox potential increased significantly as expected with the increase in the concentration of oxidized iron.

These results are apparently contrary to those obtained in previous experiments with other materials, where conductivity increases as sulfur is oxidized in addition to iron, generating sulfates, which are responsible for the high conductivity of AMD-affected rivers in the IPB [27,28]. In the steel test tubes used, the presence of sulfur is negligible—less than 0.02%—[29,30], so sulfates are barely generated through bacterial catalysis. On the other hand, the steel bars seem to favor the formation of precipitates of ferric iron salts as previously mentioned. These reasons may explain the decrease in conductivity since sulfates are not produced and oxidized iron is partially removed from the medium in the form of precipitates. Regarding the inverse relationship between changes in pH and redox potential, this is a common situation in waters affected by AMD, as demonstrated in numerous studies carried out in AMD-affected watercourses in the FPI [31–33]. However, the steel samples were hardly affected by the changes observed in the medium. It is likely that the precipitates of ferric oxihydroxides completely covering the surface of the steel acted as a passivation layer that prevented the expected corrosion.

This situation changed significantly when, in a new set of experiments, the steel samples were exposed to a culture medium to which elemental sulfur was added. The results showed a strong visible deterioration of the surface of the steel samples, indicating a biocorrosion process that involved the generation of hydrogen sulfide. The irregular profile of the evolution of physicochemical parameters, such as the simultaneous presence of yellowish precipitates (likely iron oxyhydroxides) and blackish precipitates (probable metal sulfides), as well as the strong odor of hydrogen sulfide, suggests that both oxidation and reduction processes are occurring simultaneously. This could be possible if it is observed that the culture flasks in the orbital shaker do not uniformly mix their contents. In fact, it can be observed that the medium at the upper end of the flask–in contact with air–is strongly agitated and therefore well-oxygenated, while the base has a lower degree of agitation and probably has a microaerophilic or even anaerobic condition. If this is the case, a vertical oxygen gradient would be established in the culture flasks, from the upper to the lower zone. When, as a control, an agitation system that generates vortices and homogeneously oxygenates the culture medium from the base to the upper end of the flask was used, only the consequences of iron oxidation were observed, without any process associated with sulfur reduction, and consequently without hydrogen sulfide generation or precipitation of metallic sulfides.

On the other hand, it would be feasible to consider that the inoculated bacterial consortium includes species of sulfate-reducing bacteria (SRB), although these are anaerobic and are located at the bottom or sediments of watercourses affected by AMD. SRBs feed using sulfate as an oxidizing agent and organic matter as a reducing agent. However, water samples from the river were collected from the most superficial and fast-flowing waters, which implies high oxygenation. If there were SRBs in the inocula, it would be expected that sulfur would exclusively serve as a nutritional source, using organic matter present in the water as an electron source (which is quite unlikely, since there is hardly any organic matter in the superficial waters of rivers contaminated by AMD in the IPB). Certainly, carbon steel contains a certain amount of carbon, but it is less than 0.02% $w/w$, so it could be disregarded

as a reducing power source. Therefore, another control experiment was carried out under the same agitation conditions as the aforementioned experiments, where bacterial inocula were grown exclusively with sulfur, without adding any source of iron. The results showed that sulfur exclusively was not able to support any type of bacterial growth.

There is a suggestive and alternative explanation for the results obtained, based on the fact that the most abundant bacteria in the acidic rivers of Huelva, *Acidithiobacillus* spp., is a genus that has numerous strains that in oxygen-rich environments have an obligate chemolithotrophic metabolism capable of growing using $Fe^{2+}$, S, or $H_2$ as electron donors. However, in oxygen-poor environments, their metabolism is that of a facultative anaerobic organism. That is: in aerobic environment *A. ferrooxidans* catalyses the dissimilatory oxidation of iron, sulfur, and hydrogen, but in anaerobic environment it catalyses the the reduction of iron and sulfur. In this case, $H_2$ can serve as an electron donor, especially in deep sediments. *A. ferrooxidans* can use elemental sulfur as an electron donor under both aerobic and anaerobic conditions, using oxygen or $Fe^{3+}$ ions as electron acceptors, respectively. Under anaerobic conditions, it forms $H_2S$ through a sulfur reductase and sulfate through a heterodisulfide reductase and ATP sulfurylase [34–36]. The so-called "dismutation" reactions of elemental sulfur under anaerobic conditions can be summarized in these equations:

Oxidation                                            Reduction

$$S_0 + 3H_2O \rightarrow SO_3^{2-} + 4e^- + 6H^+ \qquad\qquad S_0 + 2e^- + 2H^+ \rightarrow H_2S$$

$$SO_3^{2-} + H_2O \rightarrow SO_4^{2-} + 2e^- + 2H^+$$

That is, in the cultivation systems used, a very complex situation can arise where bacterial catalysis can simultaneously have an oxidizing character (producing $Fe^{3+}$ and $SO_4^{2-}$) or a reducing character (generating $Fe^{2+}$ and $H_2S$) depending on the concentration of oxygen in the considered microenvironment (Figure 5).

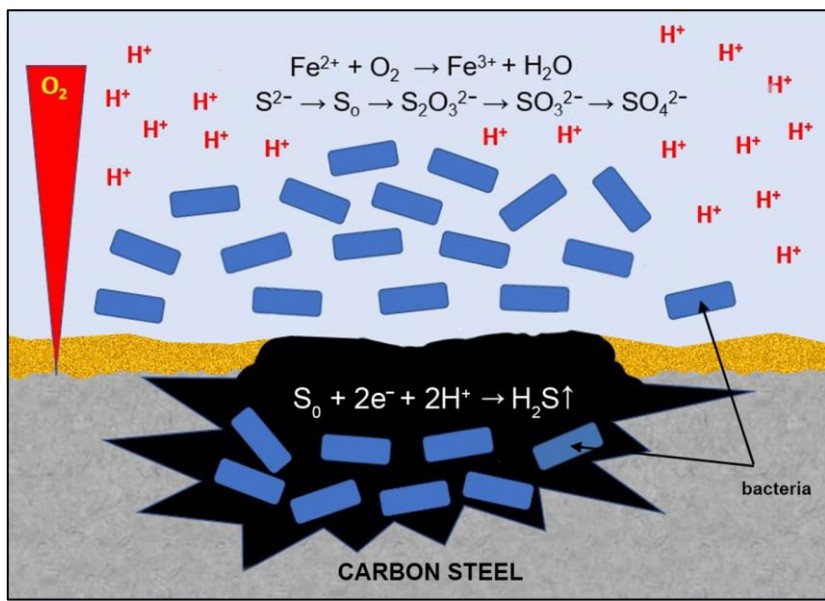

**Figure 5.** Graphical abstract of bacterial-catalyzed reactions as a function of oxygen concentration in the medium. Note that in highly oxygenated zones, oxidation processes of Fe and S occur, whereas in anaerobic or microaerophilic zones, elemental sulfur can be reduced to hydrogen sulfide, a highly corrosive acidic gas which explains the conspicuous damages of the steel bars.

## 5. Conclusions

In summary, based on various publications on the complex metabolism of *Acidithiobacillus ferrooxidans*, at the beginning of the experiment, where the steel samples are incubated in the presence of aerobic chemolithotrophic bacteria, the oxidation of iron and sulfur would occur. However, since there is presumably a vertical gradient of oxygen in the culture flasks, these oxidation reactions occur in the zones in contact with air, the most superficial ones

in the incubation medium. In the bottom of the flasks, where there is hardly any oxygen, reduction reactions of sulfates or of sulfur itself occur, resulting in the generation of $H_2S$ and metal sulfides after combining sulfides with leached metals–especially iron–from the steel bars. Hydrogen sulfide is highly corrosive [37–39], which would explain the deep damage observed in the steel samples. Figure 5 summarizes the proposed hypothesis.

The results presented in this work are preliminary in nature and therefore not without limitations. Currently, in order to test the suggested hypothesis, new experiments are being conducted that include the isolation and identification of bacterial species present in the original sample, as well as potential temporal changes in the relative composition of the bacterial populations that define the consortium used as inoculum during the experimental period. The composition and speciation (oxidized and reduced forms) of chemical elements, especially iron and sulfur, present in the culture medium throughout the experimental period will be determined. At the same time, mineralogical characterization of the precipitates will be carried out, as their composition will be indicative of redox reactions caused by a variety of bacteria presumably present in the original sample.

**Author Contributions:** Conceptualization, methodology, resources, F.C. and A.M.S.; investigation, formal analysis, F.C.; writing—original draft preparation, F.C.; writing—review and editing, F.C and A.M.S.; supervision, F.C. and A.M.S.; project administration and funding acquisition, A.M.S. All authors have read and agreed to the published version of the manuscript.

**Funding:** This research was funded by Ministry of Science and Innovation (Spain), grant number PID2021-123130OBI00.

**Data Availability Statement:** Not applicable.

**Acknowledgments:** We thank J. C. Fortes (University of Huelva, Spain) for providing us with the carbon steel used in the experiments of this work and P. Maraver for her help in the laboratory.

**Conflicts of Interest:** The authors declare no conflict of interest.

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
