# Peer review of "Biocorrosion of Carbon Steel under Controlled Laboratory Conditions"

_minerals, doi:10.3390/min13050598_

Round 1

Reviewer 1 Report

Overall, the manuscript is well-written with a flowing text. However, the authors should strive for greater clarity in their communication by providing explicit and clear descriptions of their methods and results. Despite previous research on biocorrosion, the topic remains of great interest and relevance, and this study presents new and valuable findings that readers can benefit from. The references cited should be more current, mostly covering the years 2000-2019. Nonetheless, there are some concerns that need to be addressed before publication:

Line 69: The correct abbreviation is A. ferrooxidans. Please make the change throughout the rest of the manuscript as well.

Line 93: Did you measured the iron concentration in the Agustin River water samples? In my opinion, it would have been preferable to determine the exact iron concentration and speciation in the water samples used, instead of taking them from the literature (I am now referring to the values shown in line 253)

Line 94: You must add a reference or indicate the composition for the 9K medium.

Line 101: What type of carbon steel test tubes did you use? Can you provide details on their composition?

Line 123: There is a spelling error “In a new experiment, In this new experiment,…”, please revise.

Line 126: There is no “Figure 11” in the manuscript. Please revise.

Line 133: What do you mean by “the non-steel test tubes”? I think a mistake has crept in and you need to correct it.

Line 134: I suggest you use another wording instead of "initial position"

Line 135: Was there any difference between the experiments where you used only steel and those where you used steel + Fe(SO4) as the iron source? What is the experimental version in which only 0.9 g of the weight of the steel tubes was lost? Please be more specific.

Line 144: Figure 2 is missing from manuscript.

Line 157: You need to add a reference here, for the composition of steel test tubes.

Line 159:  Please write in italics “Acidithiobacillus ferrooxidans”

Line 178: It is possible that you meant to refer to Figure 2 and not to Figure 3?

Line 181: You misspelled “culures.”

Line 189: In Figure 4A, the weight loss of the steel samples is shown as %, but previously (line 135) you expressed it in grams. Please decide to use only one form and be consistent throughout the manuscript.

Line 191: What is the magnification in Figure 4B? How were the pictures taken and with what? This is not mentioned in the Materials and Methods section.

Line 194: Can you describe the method used to remove the precipitates from the surface of the samples? Also, is there a set of "before" images of the steel samples for comparison with the "after" images available?

Line 196: Can you provide some phase contrast microscopic images of the bacterial samples?

Line 236: Both A. ferrooxidans and L. ferrooxidans are chemolithoautotrophic bacteria.

Line 241: What do you mean by “some of these species may be facultative, such as Ac. ferrooxidans”? Please explain.

Line 247: Please replace the word “necessary” with a more appropriate one. I suggest “involved”.

Line 248: Please add a reference for this statement: “The contribution of chemical oxidation through atmospheric oxygen is practically negligible at pH below 3.0.”.

Line 257: What do you mean by” …would hardly be significant”? Please elaborate.

Line 258: The type of culture you are referring to is called enrichment culture. Please use this term and add the appropriate references.

Line 278: Please add some references regarding the AMD-affected watercourses in the FPI

Line 300: Why wasn't a bacterial quantification analysis performed to quantify the main bacterial types, such as heterotrophic, iron-oxidizing, and sulphate-reducing bacteria, in both the original water samples and the following experiments? Conducting this analysis would have offered a more comprehensive understanding of the bacterial species present and the oxidation-reduction mechanisms that occurred, thereby eliminating any assumptions.

Author Response

We appreciate the detailed review work, which has significantly improved the original manuscript.

(The new line numbers correspond to the revised version)

Line 69: The correct abbreviation is A. ferrooxidans. Please make the change throughout the rest of the manuscript as well.

Line 159:  Please write in italics “Acidithiobacillus ferrooxidans”

Answer: The abbreviation for A. ferrooxidans has been corrected throughout the manuscript and is consistently written in italics.

Line 94: You must add a reference or indicate the composition for the 9K medium.

Answer: Line 106: The reference to the composition of the 9K medium was already included: #19.

Line 101: What type of carbon steel test tubes did you use? Can you provide details on their composition?

Line 157: You need to add a reference here, for the composition of steel test tubes.

Answer: Lines 112-114: The composition of the carbon steel used is shown. Composition was provided by the supplier, as indicated in Materials and Methods.

Line 123: There is a spelling error “In a new experiment, In this new experiment,…”, please revise.

Answer: Line 137: The repetition of "In this new experiment" has been corrected.

Line 133: What do you mean by “the non-steel test tubes”? I think a mistake has crept in and you need to correct it.

Answer: Line 146: The expression "non-steel test tubes" has been replaced by "steel-test tubes".

Line 134: I suggest you use another wording instead of "initial position"

Line 135: Was there any difference between the experiments where you used only steel and those where you used steel + Fe(SO4) as the iron source? What is the experimental version in which only 0.9 g of the weight of the steel tubes was lost? Please be more specific.

Line 189: In Figure 4A, the weight loss of the steel samples is shown as %, but previously (line 135) you expressed it in grams. Please decide to use only one form and be consistent throughout the manuscript.

Answer: Lines 147-149: The weight loss of the steel depending on the experimental design has been clearly indicated, and the expression "initial position" has been removed. The percentage (%) has been used throughout the manuscript regarding the expression of weight loss.

Line 144: Figure 2 is missing from manuscript.

Answer: Figure 2 was included in the original manuscript, although it has been reviewed again along with its citation in the text.

Line 126: There is no “Figure 11” in the manuscript. Please revise.

Answer: Line 137-139: The reference to Figure 11 has been removed. The reference to the experiments with steel + Fe(SO4) has also been removed, as no results had been provided due to the absence of differences with tubes that included only steel.

Line 178: It is possible that you meant to refer to Figure 2 and not to Figure 3?

Answer: Lines 193-194: References to Figures 2 and 3 have been differentiated.

Line 181: You misspelled “culures.”

Answer: Line 197: The word "culture" has been corrected.

Line 93: Did you measured the iron concentration in the Agustin River water samples? In my opinion, it would have been preferable to determine the exact iron concentration and speciation in the water samples used, instead of taking them from the literature (I am now referring to the values shown in line 253)

Answer: Lines 287-290: The exact concentration of iron and the percentage of oxidized iron in the samples used have been included.

Line 191: What is the magnification in Figure 4B? How were the pictures taken and with what? This is not mentioned in the Materials and Methods section.

Line 194: Can you describe the method used to remove the precipitates from the surface of the samples? Also, is there a set of "before" images of the steel samples for comparison with the "after" images available?

Line 196: Can you provide some phase contrast microscopic images of the bacterial samples?

Answer: The figure 4 has been expanded to include a microscopy image of the bacteria (Fig. 4A) and a photograph of the precipitates on the steel (4C, top). A scale has been added (4C, bottom).

Lines 203-209: The removal of the precipitates from the surface of the steel is described in detail. No photos of the steel before the experiment have been included because they do not show appreciable differences with the photograph shown in 4C (bottom, left).

Line 236: Both A. ferrooxidans and L. ferrooxidans are chemolithoautotrophic bacteria.

Answer: Line 262-263: The initial version already indicated that both A. ferrooxidans and L. ferrooxidans are chemolithotrophic bacteria ("aerobic chemolithotrophic bacteria, such as Acidithiobacillus ferrooxidans and Leptospirillum ferrooxidans").

Line 247: Please replace the word “necessary” with a more appropriate one. I suggest “involved”.

Answer: Line 271: The word “necessary” has been replaced by “involved”.

Line 248: Please add a reference for this statement: “The contribution of chemical oxidation through atmospheric oxygen is practically negligible at pH below 3.0.”

Answer: Lines 273-283: The sentence "The contribution of chemical oxidation through atmospheric oxygen is practically negligible at pH below 3.0" has been removed and replaced with a detailed explanation (including references) of the importance of iron(III) oxidation compared to oxygen oxidation at pH<3.

Line 257: What do you mean by” …would hardly be significant”? Please elaborate.

Answer: Line 293-295. We have added a sentence that clarifies the use of the word 'hardly'

Line 258: The type of culture you are referring to is called enrichment culture. Please use this term and add the appropriate references.

Answer: Line 293: The term "enrichment" and a reference have been used.

Line 278: Please add some references regarding the AMD-affected watercourses in the FPI

Answer: Line 318: Three new references related to "AMD-affected watercourses in the FPI" have been added.

Line 241: What do you mean by “some of these species may be facultative, such as Ac. ferrooxidans”? Please explain.

Answer: Lines 360-362: The concept of facultative metabolism referring to A. ferrooxidans is explained.

Line 300: Why wasn't a bacterial quantification analysis performed to quantify the main bacterial types, such as heterotrophic, iron-oxidizing, and sulphate-reducing bacteria, in both the original water samples and the following experiments? Conducting this analysis would have offered a more comprehensive understanding of the bacterial species present and the oxidation-reduction mechanisms that occurred, thereby eliminating any assumptions.

Answer: Lines 396-406: The limitations of the work carried out have been described in the Conclusions section, indicating that these are preliminary results and that new experiments are currently being carried out to (a) identify bacterial species, (b) analyze changes in the elemental composition of the culture medium, and (c) characterize the observed precipitates mineralogically.

Reviewer 2 Report

The manuscript presents an interesting study about the biocorrosion activity of bacterial consortia collected from waters affected by Acid Mine Drainage on carbon steel test tubes. However, the paper needs minor revisions before it is processed further, some comments follow:

 Introduction

The introduction section must contain a complex state of the art. Please introduce more studies regarding this subject and discuss them. Also, highlight the novelty of your study.

Materials and methods

Please introduce the chemical composition of the carbon steel studied. Also, details regarding the material and his role are missing.

Conclusions

The conclusion section must be improved. Add some suggestions and limitations. 

Author Response

We appreciate the detailed review work, which has significantly improved the original manuscript.

(The new line numbers correspond to the revised version)

The introduction section must contain a complex state of the art. Please introduce more studies regarding this subject and discuss them. Also, highlight the novelty of your study.

Answer: Introduction: In order to avoid extending the Introduction too much, more studies related to AMD in the IPB have been included and discussed in the Discussion section, both regarding the role of iron as an oxidant in acidic waters (lines 273-283) and in general, with the addition of new references (line 318). A paragraph has been added indicating the most important novelty of the study (lines 94-98).

Please introduce the chemical composition of the carbon steel studied. Also, details regarding the material and his role are missing.

Answer: Material and methods:  the composition of the steel has been provided in lines 112-114.

The conclusion section must be improved. Add some suggestions and limitations. 

Answer: Conclusions: Lastly the limitations of the study have been added and ongoing research has been discussed in lines 396-406.
